# Biomarkers: Promising Tools Towards the Diagnosis, Prognosis, and Treatment of Myopia

**DOI:** 10.3390/jcm13226754

**Published:** 2024-11-10

**Authors:** Varis Ruamviboonsuk, Carla Lanca, Andrzej Grzybowski

**Affiliations:** 1Department of Ophthalmology, King Chulalongkorn Memorial Hospital, Bangkok 10330, Thailand; varis.ruam@gmail.com; 2Escola Superior de Tecnologia da Saúde de Lisboa (ESTeSL), Instituto Politécnico de Lisboa, 1990-096 Lisboa, Portugal; carla.rita.costa@gmail.com; 3Comprehensive Health Research Center (CHRC), Escola Nacional de Saúde Pública, Universidade Nova de Lisboa, 1600-560 Lisboa, Portugal; 4Institute for Research in Ophthalmology, Foundation for Ophthalmology Development, 60-836 Poznan, Poland

**Keywords:** myopia, biomarkers, imaging, metabolites, genes

## Abstract

The prevalence of myopia, especially high myopia, continues to increase in several parts of the world. Thus, the use of biomarkers for early myopia detection would be highly valuable for clinical practice aiding in the prevention and myopia control management. The identification of biomarkers that can predict the risk of myopia development, severity of myopia, and treatment response are of paramount significance. In this review, we present the current state of research on biomarkers and myopia, highlighting the challenges and opportunities in biomarkers research for myopia. Although myopia biomarkers may have a role as early indicators of myopia or treatment response, the adoption of biomarkers into myopia clinical practice may only be done when tests have high accuracy, are easily measurable, minimally invasive, and acceptable to parents, children, and eye care professionals. Large consortia studies are necessary to validate biomarkers and translate evidence into clinical practice.

## 1. Introduction

The number of people globally affected by myopia has increased from 1.41 billion in 2000 to 2.62 billion in 2020, representing a significant public health concern. It is forecasted that myopia will affect 49.8% of the world’s population by 2050 [1]. The highest rates of myopia have been observed in East and Southeast Asia, where approximately 80–90% of high school children have myopia [2]. Western countries have a lower prevalence of about 20–30%. A previous study estimated that between 27 and 43% of uncorrectable visual impairment in the US population in 2050 will be directly attributable to myopia [3].

Myopia progression is associated with anatomical changes, such as axial length elongation, steep retina around the posterior pole, and retinal thinning [4]. Progression to high myopia increases the risk of development of pathologic myopia and visual impairment if left untreated, but even patients with low and moderate myopia may be at risk of developing complications [5]. Myopia complications include myopic macular degeneration (MMD), retinal detachment, cataract, and open-angle glaucoma [6]. Given the high prevalence of myopia, a significant burden on eye services is expected, especially in surgical retina services, due to the potential increased incidence of retinal detachment. Thus, there is a growing interest in identifying early biomarkers that can be used to diagnose and potentially predict myopia progression.

Myopia control treatments available include low-dose atropine, orthokeratology, anti-myopic glasses and contact lenses, and red-light therapy [7,8]. Additionally, spending time outdoors is an important strategy to delay myopia development [9]. Potentially, biomarkers can give information on the level of exposure to environmental factors, risk prediction, disease severity, and indicators of response to treatment. These promising tools may be important for the development myopia prevention strategies or new treatments as biomarkers may be indicators of myopia progression and treatment response to myopia control therapies. High-resolution imaging can also be used to detect retinal pathology features and identify early signs of choroidal atrophy or potential signs of neovascularization [10]. Optical coherence tomography (OCT) can provide useful data for the treatment and management of myopic choroidal neovascularization [11]. Biomarkers may provide valuable insights into the underlying mechanisms of myopia development [10,12,13]. Therefore, this manuscript aims to review the current state of research on biomarkers and myopia, highlighting the challenges and opportunities in biomarker research for myopia and discussing the future directions in this field.

## 2. Methods

We conducted a literature search using Pubmed and Scopus databases to identify articles related to biomarkers and myopia from January 2011 to September 2024. We used the search terms “biomarkers” AND “myopia” to retrieve relevant articles. We excluded review articles, systematic reviews and meta-analyses, animal studies, articles that were not available in English, and articles unrelated to biomarkers research. After title and abstract screening, 67 out of 284 articles were included in this review.

## 3. Anatomical Biomarkers

Multiple imaging modalities have been used to study the association between the degree of myopia and anatomical markers. Potential anatomical biomarkers in myopia and pathologic myopia are summarized in Table 1 and Table 2, respectively.

### 3.1. Myopia and High Myopia

Several studies were conducted, including patients with different levels of myopia, such as myopia and high myopia. Dong et al. found that myopic eyes had smaller foveal avascular zone area and perimeter, thicker central retinal thickness, thinner central choroidal thickness, decreased choroidal vascularity index (CVI) and choroidal vessel volume than emmetropic eyes. Additionally, high myopic eyes were found to have a larger circularity index (CI) and fractal dimension than emmetropic eyes [14]. Zhao et al. found no significant association between the flow velocity of the central retinal artery (CRA) and axial length (AL) in myopes, suggesting that AL elongation may not affect blood flow in the CRA. Furthermore, vascular resistance was not affected by AL elongation, indicating that blood flow may be similar in low and high myopia [28]. Moreover, Jiang et al. found that children with early-onset high myopia had thinner choroidal thickness (CT) than healthy controls [17]. Similarly, patients with high myopia had significantly thinner subfoveal CT and decreased choriocapillaris vessel density [19], which aligned with other studies.

Kobia-Acquah et al. conducted a study in European myopic children and evaluated longitudinal changes of CT in 250 children [18]. The authors found that CT was thickest at the perifoveal superior region and thinnest at the perifoveal nasal region. In multiple logistic regression, thinner CT was consistently associated with longer AL and higher myopic SE.

Sung et al. investigated the relationship between parapapillary atrophy (PPA) and optic nerve head (ONH) changes in 89 highly myopic eyes using spectral-domain optical coherence tomography (SD-OCT). There was a significant positive correlation between beta-PPA without Bruch’s membrane and with Bruch’s membrane opening. However, there was no association between the width of PPA and Bruch’s membrane [29].

Xu et al. studied vascular changes in patients with low-to-moderate myopia and high myopia by using OCT angiography (OCTA). The results showed that patients with high myopia had a larger flow area of the outer retina and thinner CT. AL was also correlated with flow area and CT [15].

Thickening of the choroid has been suggested as a precursor of AL growth slowing, reducing myopia progression. In a previous clinical trial low-dose atropine induced choroidal thickening, which was associated with slower myopic progression and AL elongation [30]. A deep learning segmentation algorithm on SS-OCT was used to measure the CT of participants in the ATLAS study [31]. The participants received atropine treatment for 2–4 years and were evaluated 10–20 years after treatment discontinuation. Eyes treated with atropine in childhood had significantly greater CT in inferior and nasal sectors on SS-OCT compared to those without atropine after adjusting for age, sex, and AL [20], suggesting that childhood atropine could affect the CT in adulthood.

Optical treatments for myopia control seem to follow the same pattern of results. In a two-year RCT, Defocus incorporated multiple segments (DIMS) increased CT in the DIMS group compared to controls [32]. Similarly, in another two-year RCT, highly aspherical lenslets were found to increase CT, especially in the first year of use [33]. Low-level red-light therapy also produced sustained CT thickening, with changes at three months being predictive of myopia control accuracy at 12 months [34]. Nevertheless, the effects of CT thickening may be more pronounced in the first treatment. For example, in a study where children wore MiSight contact lenses, choroidal thickening increased in the intervention group in the first year of treatment [35]. In a two-year prospective study, orthokeratology (ortho-k) lenses increased CT, but the effect diminished over time [36]. In a study by Lee et al., 153 patients were randomized into two groups: two-year use of 0.01% atropine eye drop or placebo and then followed for one year after that. CT in both treatment and control groups thickened by 12–14 μm, and during the 1-year washout phase, CT continued to thicken by 6.6 μm, significantly more than the atropine group, which showed no changes. CVI decreased in both groups during the two-year treatment phase but increased in the placebo group during the one-year follow-up after treatment cessation [37]. Thus, CT may be a useful biomarker associated with response to treatment, aiding in the identification of responders and non-responders to myopia control treatment.

### 3.2. Pathologic Myopia, Myopic Choroidal Neovascularization, and Myopic Traction Maculopathy

Wang et al. reported that CT, luminal area (LA), stromal area (SA), and CVI were lower in pathologic myopia than in emmetropia/low myopia and simple high myopia. Additionally, CT, LA, and SA were lower in simple high myopia than in emmetropia/low myopia and were negatively correlated with AL. These findings support that choroidal thinning and vascular changes influence the development of pathologic myopia [21].

For myopic maculopathy, most studies aimed to explore the biomarkers for predicting disease progression and prognosis using different imaging modalities. Yokoi et al. conducted a case series of 56 eyes from 29 myopic patients with a follow-up period exceeding 20 years to explore the prevalence of diffuse chorioretinal atrophy in myopia, specifically myopic maculopathy [26]. The study revealed that 31 eyes (55%) exhibited diffuse chorioretinal atrophy during the final visit, with 25 of these eyes demonstrating it since the initial visit. In the same study, by utilizing the META-PM study definition, 35 eyes were classified as having pathologic myopia in adulthood, with 29 of these eyes demonstrating peripapillary diffuse chorioretinal atrophy at the initial visit [26]. Additionally, Wang et al. conducted a descriptive study that evaluated patients with moderate myopic maculopathy, both with and without lacquer cracks. Eyes with lacquer cracks had significantly lower best-corrected visual acuity, higher axial length, and thinner macular and subfoveal CT than those without lacquer cracks [27].

Bontzos et al. investigated myopic choroidal neovascularization (mCNV), and their findings suggested that the progression of maculopathy in mCNV patients was correlated with thinner CT but was not correlated with central retinal thickness. Additionally, patients exhibiting any form of atrophy, whether diffuse or patchy, showed an enlargement of areas affected during the five-year follow-up period [22]. Wang et al. found that patients with mCNV who received anti-VEGF had a decrease in mean values of mCNV area, fractal dimension, vessel area, vessel length, vessel junction, junction density, and central retinal thickness. An increase in vessel density, vessel diameter, and vessel tortuosity was also found. Vessel junction was found to differ most from baseline across all anatomical measurements (mean decrease = 50.36%) [38]. Lee et al. also found that patients with CNV with subretinal hyperreflective material at baseline had better visual improvement after treatment with an anti-VEGF agent than those without subretinal hyperreflective material at six months [11]. Another study by Gao et al. revealed that the number of eyes with subretinal fluid (SRF) detected on OCT in patients with multifocal choroiditis was significantly greater than those with mCNV [39]. A study by Mularoni et al. evaluated the effectiveness of structural OCT in differentiating macular hemorrhages caused by mCNV from idiopathic macular hemorrhages in myopic patients. The researchers introduced the “myopic 2 binary reflective sign”, which is a homogeneous hyperreflective lesion (of varying sizes) in the outer retina, corresponding to the hemorrhage and a hyperreflective line separating the hemorrhage from the RPE, as a new OCT biomarker to aid in this differentiation. Results showed that using this sign in SD-OCT had 100% sensitivity and 97% specificity in distinguishing idiopathic macular hemorrhage from mCNV [23].

Wu et al. studied myopic children receiving ortho-k and assessed early changes of choroidal vasculature one month after starting ortho-k. From 50 participants, at one month, there were no significant differences in myopic progression. However, LA, SA, total choroidal area, and subfoveal choroidal thickness were significantly increased. Further analysis showed that the area under the curve (AUC) was 0.87 in a prediction model to distinguish children with slow and fast ocular elongation, including baseline CVI, one-month subfoveal choroidal thickness change, age, and sex [17].

In a study by Fang et al., myopic traction maculopathy (MTM) progression rate increased with the increasing severity of retinoschisis. MTM progression was significantly correlated with the presence of internal limiting membrane detachment and macular retinoschisis at baseline. These findings suggest that SD-OCT may be a useful tool for monitoring the progression of MTM [24]. Moreover, in a study by Park et al., the foveal curvature in the MTM group was significantly greater than in the mCNV and control groups. At the same time, there was no difference in foveal curvature between mCNV and controls [25]. Another study by Wang et al. revealed that patients with MTM had significantly lower subfoveal choroidal thickness, smaller inner retinal volume, larger outer retinal volume, and larger foveal avascular zone than patients with epiretinal membrane [40]. Similar results were reported by Wang et al. comparing patients with MTM and retinoschisis to patients with high myopia. Anatomical parameters in the MTM group were significantly different from those in the high myopia group, except for inner retinal volume, which was not different between the MTM and high myopia groups [41].

Lin et al. investigated the association between persistent Bergmeister’s papilla (PBP), a consequence of incomplete resorption of fetal vasculature, and myopia using SS-OCT. This study included participants with SE of −0.50 D or below. The researchers found that patients with both myopia and PBP had significantly lower AL and less severe myopia than those without PBP [42].

These findings indicate that choroidal thinning, vascular changes, and specific imaging biomarkers like choroidal thickness, chorioretinal atrophy, and macular structural alterations are critical in understanding pathologic myopia, myopic maculopathy, mCNV, and MTM. Advanced imaging methods, particularly OCT, play a key role in differentiating types of myopic damage and predicting progression, facilitating better clinical outcomes.

## 4. Biochemical Biomarkers

Biochemical biomarkers include microRNA expression cytokine levels in various samples, such as aqueous humor, vitreous humor, serum and vitamin levels, and myopia. The summary of potential biochemical biomarkers associated with myopia is shown in Table 3.

### 4.1. Serum Vitamin D Level

The relationship between serum vitamin D levels and myopia has been investigated in several studies. Serum vitamin D levels were not associated with myopia in children from the UK [58], young Australian adults [59], and Taiwanese children [60]. However, longer time outdoors remained a protective factor of myopia, as the myopia group had a significantly lower average time spent outdoors than the non-myopia group [58,60]. On the contrary, in a study by Jung et al., every 1 ng/mL increase in serum 25(OH)D level was associated with 0.01 D decrease in myopic refractive error, and in an adjusted model with other associated factors, including sun exposure time, the adjusted odds of myopia decreased as 25(OH)D increased [61]. Yazar et al. observed that myopia decreased with every 10 nmol/L increase in serum 25(OH)D3 concentration, but this effect was not significant in participants with Northern European backgrounds. However, East Asian ethnicity was associated with lower serum 25(OH)D3 levels and a greater prevalence of myopia [62]. The data on vitamin D and myopia are still mixed, as serum vitamin D level is associated with time outdoors, a known protective factor of myopia.

### 4.2. Metabolites

Metabolites in various sources, such as serum, aqueous humor, vitreous humor, or tear, were explored for their association with myopia and pathologic myopia. Long et al. investigated the relationship between serum inflammatory biomarkers and mCNV in patients with pathologic myopia (*n* = 63) and emmetropic age- and sex-matched participants as controls (*n* = 51). Serum hs-CRP, C3, and CH50 were significantly higher among patients with pathologic myopia compared with controls. Further binary logistic regression analysis revealed that higher serum C3 levels and older age were risk factors for mCNV [52]. Eight serum metabolites (alanine, mannose, itaconic acid, aconitic acid, O-acetylserine 1, phthalic acid, abietic acid, and salicin) were decreased in the high myopia group, while twelve serum metabolites (citric acid, aminomalonic acid, palmitoleic acid, conduritol b epoxide, shikimic acid, 4-hydroxyphenylacetic acid, hesperitin, anandamide, oxalacetic acid, pimelic acid, 2-ketoadipate and N-ethylmaleamic acid) showed increased levels in high myopes compared to the mild myopes in an age and sex-matched case-control study. In the same study, all the identified metabolites showed potential to distinguish high myopia and mild myopia with AUC values ranging from 0.59–0.71. However, the study’s limitations include the lack of adjustment to other potential risk factors [63]. In another study, the results showed significant differences in nine serum metabolites between patients with myopia and healthy controls [43]. In addition, these metabolites were confirmed in another group of patients with myopia, and all showed an AUC greater than 0.7. Two serum metabolites, γ-glutamyltyrosine, and 12-oxo-20-trihydroxy-leukotriene B4, demonstrated decent predictability of disease status with an AUC greater than 0.8. These findings suggest that serum metabolites may serve as potential biomarkers for myopia.

Nam et al. found a significant trend towards decreased myopia with higher urinary cotinine levels in 1,139 adolescents aged 12 to 18 years old. The study also showed that the risk of myopia increased as the urinary cotinine level decreased [64].

Vitreous humor was also investigated. The levels of Dickkopf 1 (DKK1) and matrix metalloproteinase 2 (MMP-2) in the vitreous humor of patients with myopia. Levels of DKK1 and MMP-2 were significantly higher in the pathologic myopia group than in the low-to-moderate myopia and control groups. These findings suggest that DKK1 and MMP-2 may play a role in the pathogenesis of myopia and could be potential therapeutic targets for myopia treatment [53]. The concentration of amino acids in aqueous and vitreous humor in patients with pathologic myopia was compared with aqueous humor samples from patients with cataracts and vitreous humor samples from patients with macular holes and epiretinal membranes. The concentration of 10 amino acids in both the aqueous humor and vitreous humor: serine, methionine, proline, creatine, lysine, arginine, tyrosine, threonine, glutamine, and asparagine were significantly higher in pathologic myopia than controls [56]. Metabolites such as D-citramalic acid, biphenyl, and isoleucylproline in the aqueous humor of patients with pathologic myopia were associated with the occurrence of subretinal fluid. D-citramalic acid, biphenyl, and isoleucylproline showed AUCs of 0.967, 0.851, and 0.801, respectively [54]. Tang et al. showed that several potential metabolites can be used as biomarkers, differentiating between pathologic myopia and myopia [55]. Shao et al. revealed that the serum transthyretin (TTR) level in the high myopia group with ocular pathology was significantly higher than that in the high myopia without ocular pathology and the control group. In addition, the TTR concentration in vitreous samples from high myopia with ocular pathology was significantly higher than that in the other two groups, and this trend was consistent with that observed in serum samples. Moreover, the study found that higher TTR concentration in vitreous samples was also associated with better post-operative visual acuity [49]. Additionally, uric acid in vitreous humor was the leading biomarker of pathologic myopia in one study with an AUC of 0.894 [55].

In aqueous humor analysis for metabolites, the levels of 8-hydroxydeoxyguanosine (8-OHdG) in myopic patients were significantly lower than in the cataract control group. This finding was positively correlated with central corneal thickness and negatively correlated with AL. Thus, oxidative stress may play a role in the development of myopia. However, there was no significant difference in the malondialdehyde level between the two groups [45]. In a study by Shao et al., 5-methoxytryptophol and cerulenin had an AUC of 0.864 and 0.879, respectively, to distinguish between AL below 24 mm and higher AL levels [44].

The metabolomics in corneal tissue of patients with myopia who underwent a small incision lenticule extraction (SMILE) refractive surgery were also studied. A previous study found that there were more than hundreds of metabolites that were different between myopia severity levels. However, azelaic acid, arginine-proline, and hypoxanthine were related to the severity of myopia. The AUC of azelaic acid, arginine-proline, and hypoxanthine were 0.982, 0.991, and 0.982, respectively [46].

Shi et al. observed that the average level of epidermal growth factor-containing fibulin-like extracellular matrix protein 1 (EFEMP1) in tears from myopic eyes was significantly lower than in emmetropic eyes. Tear concentrations of EFEMP1 were negatively correlated with SE and AL [47].

In a proteomic analysis by Wen et al., vitreous humor samples were collected from patients with other retinal diseases, such as epiretinal membrane and macular hole, with and without pathologic myopia. The group without pathologic myopia was used as the control group. The results from a small cohort of 10 patients showed expression of eukaryotic translation elongation factor 1 alpha 1 (EEF1A1), tubulin alpha 1a (TUBA1A), annexin A4 (ANXA4), myosin-9 (MYH9) and 14-3-3 protein zeta/delta (YWHAZ) were significantly greater than those in the control group. In contrast, expression of GDNF family receptor alpha-2 (GFRA2), testican-2 (SPOCK2), receptor-type tyrosine-protein phosphatase delta (PTPRD), xylosyltransferase 1 (XYLT1), and versican core protein (VCAN) was significantly lower than the control group [65].

Further studies on the role of metabolites may be important to determine the mechanisms and elucidate the pathophysiology of myopia.

### 4.3. Cytokines

There were only two studies on cytokine levels in the aqueous humor of patients with myopia. Tang et al. investigated the differences in cytokine levels in the aqueous humor of patients with high myopia compared to a cataract control group before cataract surgery. They reported an upregulation of IL-13 and downregulation of IL-15 in the high myopia group compared with the control group [50]. In another study, Shchuko et al. examined the aqueous humor of patients with mCNV and early cataracts. Their findings indicated a significant increase in the levels of inflammatory cytokines, including IL-2, IL-15, IL-17A, and TNF-alpha, in the mCNV group compared to controls. Additionally, they observed increased anti-inflammatory cytokines such as IL-5 and IL-13 and chemokines like IL-8 and RANTES. Interestingly, the VEGF level was significantly lower in the mCNV group. Furthermore, the authors reported a significant inverse correlation between myopia degree and both VEGF and IL-6 levels [57].

Tear inflammatory cytokines in patients with myopia were also analyzed in several studies. Tear samples from 132 high myopic eyes and 105 emmetropic eyes. IL-6 and MCP-1 levels were significantly higher in high myopic eyes than in controls. Further analysis showed that IL-6 and MCP-1 were able to predict myopic macular degeneration with AUC of 0.783 and 0.682, respectively [51]. Wan et al. also studied tear cytokine levels among patients with myopia, high myopia, and emmetropia. From 137 eyes, VEGFA, VEGFC, and PlGF levels in patients with high myopia were significantly less than emmetropes. At the same time, only VEGFA and PlGF levels in patients with myopia were significantly less than emmetropes [19]. In a study by Zhao et al., tear cytokines from myopic patients who underwent femtosecond laser-assisted in situ keratomileusis (FS-LASIK) were studied. After a 12-month post-operative follow-up, IL-1 beta, IL-17A, TNF-alpha, and substance P tear levels were significantly increased at 1, 3, 6, and 12-month follow-up visits compared to preoperative levels [48]. Changes in these cytokine levels may serve as indicators for myopic disease progression, especially after interventions like LASIK, and may also give some insight into subclinical changes.

## 5. Genetic Biomarkers

Most genetic biomarkers examined in past studies have included gene expression, single nucleotide polymorphisms (SNPs), and RNA associations related to myopia. The summary of potential genetic biomarkers associated with myopia is shown in Table 4. Zhang et al. explored the gene expression in individuals with myopia compared with those with normal cornea to find the possible pathogenesis of myopia. In this study, the authors used data from the Gene Expression Omnibus (GEO) database to evaluate the gene expression between these two groups. The results revealed that *NR1D1*, *PPP1R18*, *PGBD2*, and *PPP1R3D* genes were associated with myopia, and these genes were potential biomarkers for myopia [66]. In another study on gene expression profile in myopia, the authors used data from the GEO database and found that B cell, CD4+ memory T cell, CD8 central memory T cell, plasmacytoid dendritic cell, Th2, Tregs were significantly higher enriched in the myopic cornea, whereas CD8+ T cell, CD4+ T central memory cell, NK T cell, and Th1 were lower enriched [67].

In a study conducted by Kunceviciene et al., the association between single nucleotide polymorphisms (SNPs) rs662702 of the *PAX6* gene and myopia was investigated. The TT genotype is the homozygous major allele for this SNP. The odds ratio of having moderate or high myopia for participants with the CT genotype was 13.6 (95%CI 2.86–64.55) versus the TT genotype. Moreover, patients with myopia had a higher level of miR-328 expression compared to healthy controls, but the expression level was not related to the genotype of the 3′UTR of the *PAX6* gene [68].

### 5.1. SNPs

SNPs enable researchers to identify predisposing genetic factors that may be associated with the disease. In a study by Li et al., two Han Chinese university student cohorts and two European university student cohorts were assessed for SNPs associated with myopia. The minor alleles of both *ZFHX1B* rs13382811 and *SNTB1* rs6469937 were individually associated with moderate to high myopia in both Han Chinese cohorts. However, in the Bonferroni correction, *SNTB1* rs6469937 was not associated with high myopia. In European cohorts, these two SNPs alone were not associated with myopia, but the combined SNPs were significantly associated with myopia [70].

This study investigated whether specific SNPs in the *COL11A1* and *COL18A1* genes are associated with high myopia (HM) in a Han Chinese population. Analyzing data from 869 HM patients and 804 controls, researchers found no significant associations after corrections. However, the G allele of rs2236475 showed a minor increased risk for HM. Overall, common polymorphisms in these genes seem unlikely to contribute significantly to HM susceptibility, though further studies are recommended to clarify their roles [75].

Another study by Zidan et al. analyzed the association between SNPs of the *IGF-1* gene (rs6214 and rs5742632) and different types of myopia in 272 Egyptian patients with myopia and 136 controls. Expression of *IGF-1* rs6214, GA and AA genotypes, and A minor allele were significantly higher in high myopia patients than in the control group. Patients with GA and AA genotypes and A allele carriers were significantly more likely to have high myopia with an odds ratio of 1.75 (95%CI 1.03–2.90), 2.8 (95%CI 1.30–6.00), and 1.8 (95%CI 1.25–2.61), respectively. However, *IGF-1* rs5742632 genotypes were not different between the case and control groups [71].

Furthermore, a study by Yang et al. investigated the association between SNPs in *MYOC*, *HGF*, *MET*, and *ACAN* genes and high myopia in 1052 Han Chinese population with high myopia and 1070 controls. The results showed that *ACAN* rs3784757 and rs1516794 T minor allele were significantly associated with high myopia as a potential protective factor with an odds ratio of 0.83 (95%CI 0.70–0.99) and 0.79 (95%CI 0.64–0.97), respectively. Moreover, in the recessive model, rs38857 and rs10215153 in the *MET* gene and rs3784757 in the *ACAN* gene were significantly associated with high myopia and showed odds ratios of 4.14 (95%CI 1.38–12.43), 5.74 (95%CI 1.27–25.95) and 0.52 (95%CI 0.28–0.97), respectively [72].

A family-based high myopia cohort was used to investigate genetic associations with myopia. A total of 530 participants were included in the study. The researchers used both qualitative and quantitative tests to determine the association of SNPs with high myopia. Based on the results of these tests, three candidate SNPs from qualitative and quantitative analysis were identified. The results showed that *UHRF1BP1L*, *PTPRR*, and *PPFIA2* were found to be associated with myopic development within the *MYP3* locus [76].

Jiao et al. assessed the association between five SNPs and moderate to high myopia in 300 and 96 university students with moderate to high myopia and 308 and 96 without refractive error from Guangzhou and Chaoshan, respectively. Minor alleles of two SNPs, rs634990 and rs524952, located in the 15q14 region, were significantly associated with high myopia in the Guangzhou group. Furthermore, both SNPs were found to be significantly associated with high myopia in the genotypic test, additive model, and dominant model [73]. More studies on SNPs in different ethnic groups are necessary to confirm the significance of certain SNPs, confirming if there are variations in the genetic distribution related to ethnicity.

### 5.2. RNAs

Several studies have investigated the role of microRNAs (miRNAs) in high myopia. Shen et al. found high expression of up to 341 miRNAs in aqueous humor of high myopic eyes. Of these miRNAs, miR-708a and miR-148 were selected as the most significant expression difference to the cataract control eyes. Further analysis of *PAX6* gene expression in the aqueous humor of high myopic eyes was modulated by miR-708a and miR-148, negatively correlated with their levels, and positively correlated with the visual acuity of the patients [69]. Similarly, elevated levels of miR-29a in the aqueous humor of high myopia patients compared to the cataract control group were found, which may play a role in the development of myopia through inhibition of type 1 collagen synthesis [77]. In contrast, Ando et al. identified differential expression of let-7c and miR-200a in the vitreous humor of high myopia patients with macular holes compared to those with macular holes alone [78]. In another study by Zhu et al., the results showed differential expression of 249 mature miRNAs in the aqueous humor of high myopia patients compared to the control group, which may regulate signaling pathways such as TNF, MAPK, PI3K-Akt, and HIF-1. These findings highlight the potential importance of microRNAs in the pathogenesis of high myopia and may have implications for future diagnostic and therapeutic approaches [79].

You et al. studied exosomal miRNAs in the vitreous humor of patients with pathologic myopia. A Weighted Gene Co-Expression Network Analysis (WGCNA) was used in the study and revealed that miR-143-3p and miR145-145-5p, related to insulin resistance pathway and were the top two miRNAs related to myopic maculopathy [74].

Only one study was focused on circular RNAs (circRNAs) in high myopia. CircRNAs in the vitreous of patients undergoing epiretinal membrane or macular hole were categorized into two groups: high myopia and control. From a total of 486 expressed circRNAs, 339 were upregulated, and 147 were downregulated compared between high myopia and control groups. Five circRNAs were then validated using qPCR. Hsa-circDicer1, hsa-circNbea, and hsa-circPank1 were significantly upregulated in the high myopia group, while hsa-circEhmt1 was downregulated [80]. Results from these studies help to understand the pathophysiology of myopia since they are related to biochemical signaling pathways.

## 6. Discussion

Biomarkers can be classified into anatomic/imaging biomarkers, proteomic biomarkers, and genetic markers. Anatomic/imaging biomarkers are based on anatomical features observed by OCT and AL measures, while proteomic biomarkers give information on cytokines and growth factors that may be linked to choroidal, scleral, and AL changes. Single biomarkers do not provide enough information on myopia progression and the risk of developing myopic macular disease. Thus, ideally, a combination of those biomarkers may be more useful (e.g., imaging plus genetic or proteomic findings) to identify a risk assessment tool with good predictive accuracy and offer personalized treatment strategies to myopia patients and better monitoring of treatment efficacy.

Myopia biomarkers may have a promising role as early indicators of myopia or treatment response in the future. In Figure 1, we forecast promising applications of myopia biomarkers (Figure 1). However, more research is still necessary as evidence for biomarkers use as early indicators for myopia or for treatment response is still in its infancy. Anatomical biomarkers have gained significant attention due to their potential role in predicting myopia onset and progression. One of the most well-known anatomical biomarkers in myopia is AL. Studies have shown that longer AL is strongly associated with the development and progression of myopia. Current research is moving to identify and differentiate normal AL growth patterns from abnormal growth leading to myopia progression to identify the best target for myopia control efficacy [81,82]. AL growth charts are now being used in clinical practice to monitor axial length progression and monitor treatment effectiveness [83,84]. Other anatomical biomarkers include changes in the shape of the cornea and thickness of the retina. Some of these biomarkers showed the potential to identify individuals who are at risk of developing myopia or are more likely to experience myopic complications, such as thinner CT that has been associated with maculopathy progression [22]. However, current evidence is still insufficient to use CT as a biomarker of treatment efficacy in clinical practice, and further research is necessary [85].

OCTA can provide useful information on blood flow in eyes with myopia. Previous research has shown that the blood flow area in the outer retina in myopic eyes was larger than in emmetropic eyes, and there was an association with longer AL [15]. Therefore, further research into blood flow might clarify its role in the development of myopia, as long AL is strongly associated with myopia. Although research on retinal circulation in myopia and vessel density is still limited, future research can elucidate the potential as a biomarker for myopia. Thus, modern OCT-based imaging is an essential tool to measure anatomic features of myopic eyes and can help predict myopia progression.

Anatomical biomarkers in myopia hold significant potential for enhancing clinical decision-making in ophthalmology. Ocular imaging techniques have enabled the measurement of these biomarkers, which are readily available in eye clinics. This presents a promising avenue for their incorporation into existing clinical practice, apart from single imaging techniques, multimodal imaging has the advantage of being a non-invasive tool. In addition, the data obtained from these biomarkers can be utilized to create a prediction model that can help forecast the severity of myopia and facilitate the development of personalized treatment plans. Such advancements in utilizing anatomical biomarkers have the potential to revolutionize the management of myopia in clinical practice.

Further research to identify new anatomical biomarkers and develop more precise methods to measure and analyze biomarkers, including advanced imaging devices coupled with artificial intelligence algorithms, may help to identify pre-clinical disease, allowing for early treatment avoiding the development of visual impairment. This could lead to the development of more effective interventions and personalized treatment for individuals with myopia. For example, MTM patients with retinoschisis need closer observation and a more aggressive treatment plan since these patients may have a higher progression of MTM [24]. These anatomical biomarkers are easier to obtain since most of them are non-invasive tools, and ophthalmologists are more familiar with these investigations, which are used abundantly in current clinical practice.

Metabolomics emerged as a promising tool in ophthalmology to screen the metabolic changes of biofluids, cells, and tissues to aid in the prediction of the physiological and pathophysiological disease process of myopia. Several advantages may arise from using these new tools for early diagnosis, promotion of targeted prevention in primary care, and personalized myopia control treatments. Numerous metabolites in serum, aqueous humor, and vitreous humor were analyzed in previous studies to determine their association with myopia. Serum vitamin D was studied by several researchers due to its association with outdoor activity. However, some studies revealed discrepant results after adjusting for time spent outdoors, a known protective factor [58,59,60]. Even though the time spent outdoors and serum vitamin D levels are correlated, the associations with myopia may not be the same. Future systematic reviews or meta-analyses might provide us with a more robust conclusion on this matter. In addition, other metabolites were mostly different from one study to another. Therefore, many potential metabolites may be associated with myopia and its complications. There may not be a single biomarker that can be used for risk prediction, but rather a combination of several biomarkers that can predict or help in treatment planning. Nevertheless, these results lacked validation, and further studies are necessary to confirm these findings. Application in clinical settings is challenging since some of the potential biomarkers might require specialized personnel and machines to measure and analyze the results, increasing the costs.

A study on corneal metabolomics by Wu et al. hinted at a possible pathogenesis of myopia and suggested possibilities of research using corneal tissue [46]. The detailed study of the corneal tissue might allow researchers to understand more about the pathogenesis of myopia and, thus, can lead to novel treatments for myopia.

Only two studies focused on cytokines in aqueous humor of patients with myopia, one in high myopia and another in mCNV. Both studies showed increased levels of IL-13, an anti-inflammatory cytokine. However, there were different results in the levels of IL-15, a pro-inflammatory cytokine, which showed an increased level in patients with mCNV while a decreased level in patients with high myopia [50,57]. In addition to increased levels of IL-15, other pro-inflammatory and anti-inflammatory cytokines were increased in those with mCNV. This may suggest that patients with choroidal neovascularization might have some degree of ongoing inflammatory process, which is consistent with other diseases with choroidal neovascularization, such as neovascular age-related macular degeneration. Thus, further studies are important to shed light on the role of cytokines used in clinical practice to detect early signs of neovascularization in patients with myopia. One study focused on tear inflammatory cytokines in high myopia and myopic macular degeneration [51]. Although the AUC of IL-6 and MCP-1 in tears to predict myopic macular degeneration were below 0.8, further studies on tears might provide more information on biomarkers and pathogenesis of myopia. Additionally, the measurement of aqueous protein might be affected by the lens status and mydriatic eye drop use prior to the collection and could be a confounder [86]. The use of biochemical biomarkers, such as inflammatory cytokines or oxidative stress markers, may provide information on underlying mechanisms of myopia and potential targets for therapeutic interventions, leading to improved diagnosis, better prediction of myopia progression, and more effective myopia control therapies. However, further research is needed to fully understand the potential of biomarkers in myopia and how they can be translated into clinical practice.

There were some interesting potential metabolites in myopia, which might allow further understanding of myopia pathogenesis and insight into myopia prevention. Two serum metabolites, γ-glutamyltyrosine, and 12-oxo-20-trihydroxy-leukotriene B4, demonstrated an AUC greater than 0.8 in predictability of disease status [43]. Therefore, this might allow ophthalmologists to use these markers to determine the disease status and have personalized treatment plans. Moreover, in an age-sex-matched case-control study by Ke et al., several serum metabolites were found that could be used to distinguish between high and mild myopia. However, AUC values varied from 0.59–0.71, which were quite low. Those results may be related to a small sample size of 80 participants in the study, and these metabolites might vary between individuals [63]. Although some studies revealed very high AUC values, the small sample size in several studies makes generalization questionable. Thus, to apply these biomarkers into clinical practice, clinicians need to know thoroughly which metabolites are valid and significant. Biomechanical biomarkers collected by using vitreous taps are unlikely to be used in a clinical setting. However, they may provide important insights related to the disease mechanisms. At present, compared to measures of refraction and AL, the use of biomechanical biomarkers seems more challenging due to the invasive nature and need for the most expensive testing tools.

There were many studies on genetic biomarkers, such as microRNA, and genetic variations, such as SNPs. Interestingly, all the studies included in this review analyzed genes, similar to studies on metabolites. Thus, many different potential genes may be associated with myopia, and further studies with larger sample sizes and various ethnicities are necessary. In clinical practice, genetic testing is helpful in single-gene disorders. However, myopia is a complex trait associated with both genetic and environmental factors and gene-environment interactions. Therefore, it may be difficult to identify a single biomarker that predicts the risk of developing myopia. Single biomarkers of myopia may have low sensitivity and specificity, and a panel of biomarkers may, in theory, lead to higher accuracy. Translating biomarkers into myopia clinical practice to help in risk prediction still has a long way to go until daily practice implementation.

The use of biomarkers for myopia prevention purposes is still far from clinical application. It is still challenging to use one biomarker to predict or assess the stage of myopia. The results of included studies were mostly on associations between various factors and myopia, and more studies on biomarkers and treatment response to new myopia control treatments are lacking. Combining multiple biomarkers for myopia could enhance diagnostic accuracy, as individual markers may only provide limited insights into the condition’s complex genetic and environmental basis. Multi-biomarker strategy may also help identify distinct subtypes of myopia, leading to more tailored interventions. However, it also requires careful validation to ensure clinical feasibility. Future research can be helpful, for example, supporting parents with a family history of myopia using genetic and biomarker testing to adjust their lifestyle or consider preventive treatment based on the results of those tests. Similarly, the use of biomarkers in myopia control trials is not common. Thus, further research is necessary to understand if biomarkers can be used as reliable tools to prevent myopia or myopia progression. In the future, it would be of paramount importance if eye care professionals could choose and select individualized treatment plans based on therapeutic and prognostic biomarkers of myopia to predict and maximize treatment efficacy and outcomes. This may all be possible in the future with the fast growth of omics technologies, though many challenges remain to be overcome. Furthermore, while identifying a promising biomarker is valuable, it would be ideal if it could be measured or analyzed using commonly available devices. This would facilitate the biomarker’s application in clinical settings. Otherwise, there could be challenges in translating it from research to clinical application.

The use of biomarkers in myopia offers several advantages, such as early detection and timely interventions, and can potentially lead to personalized treatments. Genetic, proteomic, and imaging markers can provide information on myopia progression and treatment responses to different myopia therapies available. Nevertheless, although many biomarkers have been identified in previous experiments and studies, these have not been validated for clinical use. Disadvantages include variable expressions in different populations and ethnicities, difficulties in isolating single biomarkers that provide high predictive value, and lack of longitudinal studies with higher follow-ups to confirm reliability and accuracy over time. Additionally, it seems that there is inconsistent reproducibility among previous studies, probably due to a lack of specificity and different testing methods. Further studies should aim to validate biomarkers using larger sample sizes and populations with more diverse racial and ethnic backgrounds, as myopia prevalence and progression are known to differ (e.g., Asian versus European populations). Some of the biomarkers may be only associated with later stages of myopia progression or in the presence of myopic macular degeneration, which hinders their use as a tool for a timely diagnosis. The interpretation of data related to biomarkers requires specialized knowledge that is not available in many of the myopia clinics.

Further research should focus on the validation of biomarkers, assessing their predictive accuracy over time, and comparing groups with myopia progression and no myopia progression. A new area of further research would be to introduce biomarker analysis in myopia control trials to assess the responses to different types of treatment, such as atropine, optical interventions, or red-light therapy. Current analysis of biomarkers relies on advanced laboratory assessments with specialized software to analyze genetic and proteomic data with high costs. In addition, invasive sampling, such as the one used to collect aqueous humor samples, may not be practical in a clinical setting and for widespread use. Further work into non-invasive biomarkers is also necessary, using imaging techniques to improve their performance to detect early structural changes. Advancements in imaging techniques coupled with AI may provide higher accuracy and affordable solutions to implement in myopia clinics in the near future.

## 7. Conclusions

Many anatomical, biochemical, and genetic biomarkers have been associated with myopia. However, the study of biomarkers in myopia is still in its early stages. Thus, implementation into clinical practice still needs further research. Future studies based on machine learning algorithms may provide guidelines for clinical practice as biochemical and genetic assessment may not be cost-effective, being suitable only for a small number of patients. Nevertheless, the identification of genetic biomarkers associated with myopia may be important, leading to earlier diagnosis and personalized treatment approaches in the future. Further research is necessary to improve the development of imaging biomarkers that can accurately predict the progression of myopia and allow for more targeted interventions, as imaging tools are abundant in eye clinics. To integrate testing on biomarkers into myopia, clinical practice tests need to have high accuracy, be easily measurable, minimally invasive, and acceptable to parents, children, and eye care professionals. Large consortia studies are necessary to validate biomarkers and translate evidence into clinical practice.

## Figures and Tables

**Figure 1 jcm-13-06754-f001:**
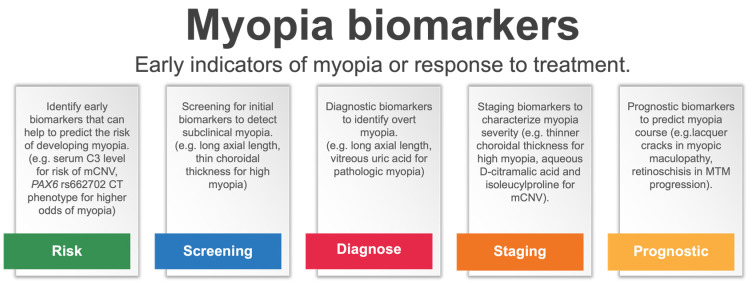
Promising applications of myopia biomarkers. mCNV, myopic choroidal neovascularization; MTM, myopic traction maculopathy.

**Table 1 jcm-13-06754-t001:** Potential anatomical biomarkers associated with myopia and high myopia.

Authors	Biomarker	Age	Sample Size	Ethnicity	Type of Myopia	Method of Detection	Results
Dong et al. (2022) [14]	FAZ, CCT, CVI, CVV	18–60	127	Chinese	Myopia (SE ≥ −10.0 D)	SS-OCT	Smaller FAZ (0.49 ± 0.12 vs. 0.35 ± 0.11 mm^2^), CCT (406.52 ± 64.78 vs. 272.25 ± 78.96 µm), CVI (0.42 ± 0.11 vs. 0.29 ± 0.08) and CVV (0.14 ± 0.04 vs. 0.07 ± 0.03 mm^3^) in high myopic eyes
Dong et al. (2022) [14]	CRT	18–60	127	Chinese	Myopia (SE ≥ −10.0 D)	SS-OCT	Thicker CRT in high myopic eyes (242.74 ± 19.42 vs. 260.21 ± 22.46 µm)
Xu et al. (2023) [15]	FA-OR, CT	18–65	74	Chinese	Myopia (SE ≤ −0.75 D)	OCTA	Larger FA-OR (1.231 ± 0.789 vs. 1.327 ± 0.662 mm^2^) and thinner CT (277.940 ± 65.099 vs. 230.272 ± 65.183 µm) in high myopia
Jiang et al. (2023) [16]	CT	<18	80	Taiwanese	Early-onset high myopia (SE ≤ −4.0 D for age ≤ 5, ≤−6.0 D for age 6–8)	SS-OCT	Thinner CT in early-onset high myopia (331.73 ± 75.23 vs. 183.96 ± 91.10 µm)
Wu et al. (2023) [17]	CVI, SFCT	8–12	50	Chinese	Myopia receiving ortho-k (SE −0.75–−5.0 D)	SS-OCT, OCTA	AUC of a prediction model achieved 0.872
Kobia-Acquah et al. (2023) [18]	CT	6–16	250	European	Myopia	SS-OCT	Thinner CT associated with longer AL and higher myopic SE.
Wan et al. (2024) [19]	CT, SFVD	18–60	137	Chinese	Myopia (SE −6.0–−0.5 D), high myopia (SE ≤ −6.0 D and/or AL ≥ 26.5 mm)	OCTA	Thinner CT and decreased SFVD in high myopia than normal.
Li et al. (2024) [20]	CT	6–12 (at initial recruitment)	211	Asian	Myopia (SE −1.0–−6.0 D and astigmatism −1.5 D or less)	SS-OCT	Thicker CT in eyes with childhood atropine use. (Mean difference: 32.1 μm, inner inferior; 23.5 μm, outer inferior; 21.8 μm, inner nasal; and 21.8 μm, outer nasal.)

AL, Axial length; AUC, Area under ROC curve; CCT, Central choroidal thickness; CRT, Central retinal thickness; CT, choroidal thickness; CVI, Choroidal vascularity index; CVV, Choroidal vessel volume; FAZ, Foveal avascular zone; FA-OR, flow area of the outer retina; OCTA, optical coherence tomography angiography; SE, spherical equivalent; SFCT, subfoveal choroidal thickness; SFVD, subfoveal choriocapillaris vessel density; SS-OCT, Swept-source optical coherence tomography.

**Table 2 jcm-13-06754-t002:** Potential anatomical biomarkers associated with pathologic myopia.

Authors	Biomarker	Age	Sample Size	Ethnicity	Type of Myopia	Method of Detection	Results
Wang et al. (2022) [21]	CT, CVI, LA, SA	21–59	80	Chinese	Pathologic myopia (SE ≤ −6.0 D or AL ≥ 26.5 mm with maculopathy)	SS-OCT	Smaller in size among pathologic myopia
Bontzos et al. (2022) [22]	CT	≥18	72	Caucasian	mCNV (SE ≤ −6.0 D and AL ≥ 26 mm with mCNV)	SD-OCT	Thinner CT correlated with progression of maculopathy in mCNV
Lee et al. (2023) [11]	SHM	N/A	43	Taiwanese	mCNV (SE ≤ −6.0 D or AL ≥ 26.5 mm with active CNV)	SD-OCT	Eyes with SHM had better visual improvement at 6 months
Mularoni et al. (2024) [23]	“Myopic 2 binary sign”	N/A	52	Caucasian	mCNV (Myopic maculopathy with new macular hemorrhage)	SD-OCT	100% sensitivity and 97% specificity for distinguishing mCNV hemorrhage from IMH using this sign
Fang et al. (2022) [24]	Retinoschisis	21–76	120	Chinese	MTM (SE ≤ −6.0 D or AL ≥ 26.5 mm with MTM)	SD-OCT	Increasing severity associated with higher MTM progression rate.
Park et al. (2019) [25]	Foveal curvature	N/A	199	Korean	MTM (SE ≤ −6.0 D or AL ≥ 26.5 mm with MTM)	SD-OCT	Greater curvature than mCNV and controls
Yokoi et al. (2016) [26]	Peripapillary diffuse chorioretinal atrophy	≤15	29	Japanese	Myopic maculopathy (SE ≤ −8.0 D or AL ≥ 26.5 mm with stage 2 or higher myopic maculopathy)	Color fundus photography	Pre-existing since childhood in pathologic myopia (29/35 eyes)
Wang et al. (2013) [27]	Lacquer cracks	N/A	69	Taiwanese	Myopic maculopathy (SE ≤ −6.0 D with chorioretinal atrophy)	SD-OCT	Associated with lower BCVA, higher AL, and thinner subfoveal CT

AL, Axial length; BCVA, best-corrected visual acuity; CRT, Central retinal thickness; CT, choroidal thickness; CVI, Choroidal vascularity index; IMH, idiopathic macular hemorrhage; LA, luminal area; mCNV, myopic choroidal neovascularization; OCTA, optical coherence tomography angiography; SA, stromal area; SD-OCT, Spectral-domain optical coherence tomography; SE, spherical equivalent; SFCT, subfoveal choroidal thickness; SHM, subretinal hyperreflective material; SS-OCT, Swept-source optical coherence tomography.

**Table 3 jcm-13-06754-t003:** Potential biochemical biomarkers associated with myopia.

Authors	Biomarker	Age	Sample Size	Ethnicity	Type of Myopia	Biological Material	Method of Detection	Results
Myopia								
Dai et al. (2019) [43]	γ-glutamyltyrosine and 12-oxo-20-trihydroxyleukotriene B4	N/A	60 (discovery), 39 (validation)	Chinese	Myopia	Serum	QTOF-MS	AUC > 0.8
Shao et al. (2023) [44]	5-methoxytryptophol and cerulenin	N/A	34	Chinese	Myopia	Aqueous humor	LC-MS	AUC > 0.8 in differentiation between axial length of less than 24 mm or higher.
Kim et al. (2016) [45]	8-OHdG	N/A	38	Korean	Myopia	Aqueous humor	ELISA	Lower level than cataract controls
Wu et al. (2023) [46]	Azelaic acid, Arg-Pro, hypoxanthine	18–45	221	Chinese	Myopia	Corneal tissue	UHPLC-MS	AUC > 0.98
Shi et al. (2023) [47]	EFEMP1	18–70	131	Chinese	Myopia	Tear	ELISA	Significantly lower concentration in myopia
Zhao et al. (2024) [48]	IL-1 beta, IL-17A, TNF-alpha, and substance P	18–45	73	Chinese	Myopia underwent FS-LASIK	Tear	Luminex bead-based multiplex assay	Increased levels at 1, 3, 6 and 12-month follow-up visit compared with pre-operative baseline
High myopia								
Shao et al. (2011) [49]	TTR concentration	≥18	202	Chinese	High myopia with ocular pathology	Serum, aqueous humor, vitreous humor	ELISA	Higher concentration than high myopia without ocular pathology
Tang et al. (2022) [50]	IL-13	N/A	40	Chinese	High myopia	Aqueous humor	Luminex bead-based multiplex assay	Higher level than cataract controls
Guo et al. (2023) [51]	IL-6, MCP-1	N/A	357	Chinese	High myopia	Tear	Bead-based cytokine antibody array	Higher levels in high myopia group. AUC of 0.783 and 0.682 for predicting MMD, respectively.
Wan et al. (2024) [19]	VEGFA, VEGFC and PlGF	18–60	137	Chinese	High myopia	Tear	ELISA	Lower levels in high myopia than emmetropia group
Pathologic myopia								
Long et al. (2013) [52]	Serum C3 level	17–70	114	Chinese	Pathologic myopia	Serum	Immune nephelometry	Higher level increases risk of mCNV
Peng et al. (2020) [53]	DKK1 and MMP-2	N/A	137	Chinese	Pathologic myopia	Vitreous humor	MILLIPLEX multiplex assay	Higher levels than low-to-moderate myopia and controls
Wei et al. (2023) [54]	D-citramalic acid, biphenyl and isoleucylproline	N/A	22	Chinese	Pathologic myopia	Aqueous humor	GC-MS, LC-MS	AUC > 0.8 in differentiation of mCNV and no mCNV
Tang et al. (2023) [55]	Uric acid	N/A	62	Chinese	Pathologic myopia	Vitreous humor	UPHLC-MS	AUC of 0.894 for differentiation between pathologic myopia and myopia
Lian et al. (2022) [56]	10 amino acids (serine, methionine, proline, creatine, lysine, arginine, tyrosine, threonine, glutamine, and asparagine)	≥18	60	Chinese	Pathologic Myopia	Aqueous humor/vitreous humor	LC-MS	Significantly higher concentration in pathologic myopia
Shchuko et al. (2017) [57]	IL-5, IL-13, IL-8	N/A	34	Russian	mCNV	Aqueous humor	Continuous-flow fluorometry with a double-beam laser automatic analyzer	Higher levels than early cataract patients.

N/A, Not available. Arg-Pro, arginine-proline; AUC, Area under ROC curve; EFEMP1, epidermal growth factor-containing fibulin-like extracellular matrix protein 1; ELISA, Enzyme-linked immunosorbent assay; GC-MS, Gas chromatography-mass spectrometry; LC-MS, Liquid chromatography-mass spectrometry; mCNV, myopic choroidal neovascularization; MMD, myopic macular degeneration; PlGF, placental growth factor; QTOF-MS, Quadrupole time-of-flight mass spectrometry; VEGF, vascular endothelial growth factor; UHPLC-MS, ultra-high-performance liquid chromatography-mass spectrometry.

**Table 4 jcm-13-06754-t004:** Potential genetic biomarkers associated with myopia.

Authors	Biomarker	Age	Sample Size	Ethnicity	Type of Myopia	Biological Material	Method of Detection	Results
Myopia								
Zhang et al. (2023) [66]	*NR1D1*, and *PPP1R18* genes	N/A	N/A	N/A	Myopia	N/A	Machine-learning bioinformatics	Downregulation in myopia
Zhang et al. (2023) [66]	*PGBD2*, and *PPP1R3D* genes	N/A	N/A	N/A	Myopia	N/A	Machine-learning bioinformatics	Upregulation in myopia
Kunceviciene et al. (2019) [68]	*PAX6* rs662702, CT genotype	18–40	451	European	Myopia	Blood	PCR	Associated with higher odds of myopia
Shen et al. (2022) [69]	miR-708a/miR-148	≥18	30	N/A	High Myopia	Aqueous humor	qRT-PCR	Upregulated expression in high myopia
High myopia								
Li et al. (2017) [70]	*ZFHX1B* rs13382811, minor T allele	N/A	800	Chinese	High myopia	Blood	PCR	Associated with higher odds ratio of high myopia
Zidan et al. (2016) [71]	*IGF-1* rs6214, GA, AA genotypes and A minor allele	N/A	408	Egyptian	High myopia	Blood	PCR	Associated with higher odds ratio of high myopia
Yang et al. (2014) [72]	*ACAN* rs3784757 and rs1516794, T minor allele	11–78	2122	Chinese	High myopia	Blood	SNaPshot	Associated with lower odds ratio of high myopia
Yang et al. (2014) [72]	*MET* rs38857 and rs10215153, homozygous minor genotypes	11–78	2122	Chinese	High myopia	Blood	SNaPshot	Associated with higher odds ratio of high myopia
Jiao et al. (2012) [73]	rs634990 and rs524952 in the 15q14 region, minor alleles	N/A	800	Chinese	High myopia	Blood	PCR	Associated with higher odds ratio of high myopia
Pathologic myopia								
You et al. (2023) [74]	miR-143-3p, miR-145-5p	N/A	27	Chinese	Pathologic myopia	Vitreous humor	PCR	Most related with myopic maculopathy

N/A, Not available. PCR, Polymerase chain reaction; qRT-PCR, quantitative reverse transcriptase-polymerase chain reaction.

## Data Availability

Data sharing is not applicable.

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
