# Peer review of "Biomarkers: Promising Tools Towards the Diagnosis, Prognosis, and Treatment of Myopia"

_jcm, 2024, doi:10.3390/jcm13226754_

Round 1
Reviewer 1 Report
Comments and Suggestions for Authors
This manuscript provides an extensive overview of biomarkers associated with myopia, highlighting their potential roles in diagnosis, prognosis, and treatment. The topic is of great relevance, given the growing prevalence of myopia globally, and the authors have done a commendable job in compiling and summarizing the latest research in this field.
However, there are several areas that need improvement:
1.The manuscript's organization could be clearer, as the transitions between sections are not always smooth. For example, the last paragraph of Section 5 seems unrelated to its title, and there is overlapping content in Sections 6, 7, and 8. Additionally, the last paragraph of Section 8 deviates from its focus on metabolic factors.
Recommendation: Reorganize the sections for clearer categorization of content. Ensure that each section has a defined focus and that transitions between paragraphs are more logical and coherent. Use introductory and concluding sentences to guide the reader through each section.
2.The manuscript provides an extensive list of studies and their findings but lacks critical commentary on the results. There is minimal discussion of the advantages and disadvantages of the different biomarkers or their clinical applicability.
3.Several important topics are not thoroughly addressed in the discussion. For example, blood flow biomarkers are briefly mentioned but not discussed in depth. Moreover, there is little emphasis on the recent advancements in imaging or detection technologies and their potential breakthroughs in identifying biomarkers.
Comments on the Quality of English LanguageOverall, the language is clear, but there are a few awkwardly structured sentences. For example, line 189, "comparing patients with MTM with retinoschisis," and lines 103-104, which lack clarity.Recommendation: Improve readability by rephrasing complex sentences and eliminating redundant phrasing. A thorough copy-editing process will help smooth out these issues and improve the flow of the text.
Author Response
Response to Reviewer 1 Comments
This manuscript provides an extensive overview of biomarkers associated with myopia, highlighting their potential roles in diagnosis, prognosis, and treatment. The topic is of great relevance, given the growing prevalence of myopia globally, and the authors have done a commendable job in compiling and summarizing the latest research in this field.
We thank the reviewer for the thorough reading of our manuscript. We appreciate all the comments.
However, there are several areas that need improvement:
Point 1: The manuscript's organization could be clearer, as the transitions between sections are not always smooth. For example, the last paragraph of Section 5 seems unrelated to its title, and there is overlapping content in Sections 6, 7, and 8. Additionally, the last paragraph of Section 8 deviates from its focus on metabolic factors.
Recommendation: Reorganize the sections for clearer categorization of content. Ensure that each section has a defined focus and that transitions between paragraphs are more logical and coherent. Use introductory and concluding sentences to guide the reader through each section.
Response 1: We have re-organized the numbering of each section. Section 6, 7 and 8 are sub-section of the same main section. We have also re-organized the paragraphs and added new sentences to improve the transitions between paragraphs.
Point 2: The manuscript provides an extensive list of studies and their findings but lacks critical commentary on the results. There is minimal discussion of the advantages and disadvantages of the different biomarkers or their clinical applicability.
Response 2: We have expanded the discussion on clinical applicability of biomarkers as suggested on lines 850-871.
Point 3: Several important topics are not thoroughly addressed in the discussion. For example, blood flow biomarkers are briefly mentioned but not discussed in depth. Moreover, there is little emphasis on the recent advancements in imaging or detection technologies and their potential breakthroughs in identifying biomarkers.
Response 3: We appreciated your suggestion. In the revised version, we have added a discussion on those topics on lines 644-645.
Comments on the Quality of English Language
Overall, the language is clear, but there are a few awkwardly structured sentences. For example, line 189, "comparing patients with MTM with retinoschisis," and lines 103-104, which lack clarity. Recommendation: Improve readability by rephrasing complex sentences and eliminating redundant phrasing. A thorough copy-editing process will help smooth out these issues and improve the flow of the text.
Response: We thank the reviewer for the comment. We have re-phrased the sentences as suggested.
Reviewer 2 Report
Comments and Suggestions for Authors
Ruamviboonsuk. V., et al discussed the prevalence of myopia, especially high myopia, is rising globally, making early detection crucial for prevention and management. Biomarkers could play a significant role in predicting myopia risk, severity, and treatment response. In this review the authors discuss the current research on myopia biomarkers, highlighting the challenges and potential in the field. However, biomarkers must be highly accurate, easily measurable, minimally invasive, and acceptable to patients and professionals before being integrated into clinical practice. Large-scale studies are needed to validate these biomarkers and bring them into clinical use.
The authors mentioned that many anatomical, biochemical, and genetic biomarkers have been linked to myopia, but research is still in its early stages, delaying their use in clinical practice. Machine learning may help develop guidelines, as biochemical and genetic assessments are costly and suitable for limited patients. Identifying genetic biomarkers could lead to earlier diagnosis and personalized treatment. Improving imaging biomarkers for prediction of myopia progression, are widely available.
The authors made a broad discussion on myopia biomarkers and their promising role as early indicators of myopia or treatment response in the future.
High appreciation goes to the authors for providing appropriate research ethical statements and the information-loaded tables.
The basic approach to writing the manuscript is good. More appreciation goes to the authors for the discussion and conclusions. The future studies in the manuscript are missing where it needs to explain the significance and importance, as well as the myopia disease progression.
Although this kind of study has been done and highly discussed very recently in the scientific community, Overall, the clarity of the text is understandable and needs some readjustments. The manuscript has minor typographical and grammatical errors.
The authors are advised to consider the comments below:
Comments:
1. Figure 1: It is a hypothetical chart of how biomarkers can be involved in myopia. It would be a lot more convincing to put examples with references.
2. Crucial references are missing like “Machine learning may help develop guidelines” – Ref: https://bmcophthalmol.biomedcentral.com/articles/10.1186/s12886-023-03119-5
3. Needs to change the writing style like “ Dong et al. found that/ Zhao et al. found/ Sung et al. investigated” – sounds like summarizing an old manuscript in 1-2 lines. It would be more convincing to write and put more thoughts inside the writing and use the paper as a reference.
Comments on the Quality of English LanguageOverall, the clarity of the text is understandable and needs some readjustments. The manuscript has minor typographical and grammatical errors.
In general, the manuscript can accomplish the caliber of quality for consideration for publication in the journal “Journal of Clinical Medicine”
Author Response
Response to Reviewer 2 Comments
Comments and Suggestions for Authors
Ruamviboonsuk. V., et al discussed the prevalence of myopia, especially high myopia, is rising globally, making early detection crucial for prevention and management. Biomarkers could play a significant role in predicting myopia risk, severity, and treatment response. In this review the authors discuss the current research on myopia biomarkers, highlighting the challenges and potential in the field. However, biomarkers must be highly accurate, easily measurable, minimally invasive, and acceptable to patients and professionals before being integrated into clinical practice. Large-scale studies are needed to validate these biomarkers and bring them into clinical use.
The authors mentioned that many anatomical, biochemical, and genetic biomarkers have been linked to myopia, but research is still in its early stages, delaying their use in clinical practice. Machine learning may help develop guidelines, as biochemical and genetic assessments are costly and suitable for limited patients. Identifying genetic biomarkers could lead to earlier diagnosis and personalized treatment. Improving imaging biomarkers for prediction of myopia progression, are widely available.
The authors made a broad discussion on myopia biomarkers and their promising role as early indicators of myopia or treatment response in the future.
High appreciation goes to the authors for providing appropriate research ethical statements and the information-loaded tables.
We thank the reviewer for the thorough reading of our manuscript. We appreciate all the comments.
The basic approach to writing the manuscript is good. More appreciation goes to the authors for the discussion and conclusions. The future studies in the manuscript are missing where it needs to explain the significance and importance, as well as the myopia disease progression.
We added more information on future studies and also studies on myopic progression as suggested on lines 707-716.
Although this kind of study has been done and highly discussed very recently in the scientific community, Overall, the clarity of the text is understandable and needs some readjustments. The manuscript has minor typographical and grammatical errors.
The authors are advised to consider the comments below:
Comments:
Comment 1: Figure 1: It is a hypothetical chart of how biomarkers can be involved in myopia. It would be a lot more convincing to put examples with references.
Response 1: We add some examples in the figure as suggested.
Comment 2: Crucial references are missing like “Machine learning may help develop guidelines” – Ref: https://bmcophthalmol.biomedcentral.com/articles/10.1186/s12886-023-03119-5
Response 2: We have included the mentioned reference in the revised manuscript under Section 5. Genetic Biomarkers.
Comment 3: Needs to change the writing style like “Dong et al. found that/ Zhao et al. found/ Sung et al. investigated” – sounds like summarizing an old manuscript in 1-2 lines. It would be more convincing to write and put more thoughts inside the writing and use the paper as a reference.
Response 3: We thank the reviewer for the comment. We have re-phrased several sentences throughout the manuscript.
Comments on the Quality of English Language
Overall, the clarity of the text is understandable and needs some readjustments. The manuscript has minor typographical and grammatical errors.
In general, the manuscript can accomplish the caliber of quality for consideration for publication in the journal “Journal of Clinical Medicine”
Response: We appreciated the comment. We have thoroughly reviewed the manuscript according to your suggestions.
Reviewer 3 Report
Comments and Suggestions for Authors
The manuscript by Ruamviboonsuk et al is a review article that attempts to collect and summarise recent data on biomarkers as promising tools for the diagnosis, prognosis and treatment of myopia. Although the authors have performed a good literature search on the topic, the resulting manuscript is descriptive and lacks analysis of the data presented. As a result, the conclusions are very broad and uninformative, suggesting a need for further research. However, further reader would like to know the answers to the following questions
- What are the advantages and/or disadvantages of already characterized biomarkers?
- What are the limitations of their use?
- Would it be possible to classify them?
- What is the potential of using their combinations and which combinations are the most promising?
- What are the most promising areas for further development?
- What are the technical and economic limitations and ease of use?
Unfortunately, the current review does not provide the answers.
Author Response
Response to Reviewer 3 Comments
The manuscript by Ruamviboonsuk et al is a review article that attempts to collect and summarise recent data on biomarkers as promising tools for the diagnosis, prognosis and treatment of myopia. Although the authors have performed a good literature search on the topic, the resulting manuscript is descriptive and lacks analysis of the data presented. As a result, the conclusions are very broad and uninformative, suggesting a need for further research. However, further reader would like to know the answers to the following questions
- What are the advantages and/or disadvantages of already characterized biomarkers?
We added the following sentences on lines 855-863.
“The use of biomarkers in myopia offers several advantages, such as early detection, timely interventions and can potentially lead to personalized treatments. Genetic, proteomic and imaging markers can provide information of myopia progression and treatment responses to different myopia therapies available. Nevertheless, although many biomarkers have been identified in previous experiments and studies, these have not been validated for clinical use. Disadvantages include variable expressions in different populations and ethnicities, difficulties in isolate single biomarkers that provide high predictive value and lack of longitudinal studies with higher follow-ups to confirm reliability and accuracy over time.”
- What are the limitations of their use?
We answered to this question on lines 863-871.
“Additionally, it seems that there is inconsistent reproducibility among previous studies probably due to lack of specificity and different testing methods. Further studies should aim to validate biomarkers using larger sample sizes and populations with more diverse racial and ethnic backgrounds has myopia prevalence and progression is known to differ (e.g. Asian versus European populations). Some of the biomarkers may be only associated with later stages of myopia progression or in the presence of myopic macular degeneration which hinders their use as a tool for a timely diagnose. The interpretation of data related with biomarkers requires specialized knowledge that is not available in many of the myopia clinics.”
- Would it be possible to classify them?
We added the answer to this answer on lines 616-620 as following.
“Biomarkers can be classified into anatomic/imaging biomarkers, proteomic biomarkers and genetic markers. Anatomic/imaging biomarkers are based on anatomical features observed by OCT and AL measures, while proteomic biomarkers give information on cytokines and growth factors that may be linked to choroidal, scleral and AL changes.”
- What is the potential of using their combinations and which combinations are the most promising?
The following sentences were added on lines 620-624.
“No single biomarker provides enough information on myopia progression and the risk of developing myopic macular disease. Thus, ideally, a combination of those biomarkers maybe more useful (e.g. imaging plus genetic or proteomic findings) to identify a risk assessment tool with good predictive accuracy and offer personalized treatment strategies to myopia patients and better monitoring of treatment efficacy.”
- What are the most promising areas for further development?
We answered to the question on lines 872-883 as following.
“Further research should focus on validation of biomarkers assessing their predictive accuracy over time comparing groups with myopia progression and no myopia progression. A new area of further research would be to introduce biomarker analysis in myopia control trials to assess the responses to different types of treatment, such as atropine, optical interventions or red-light therapy. Current analysis of biomarkers relies on advanced laboratory assessed with specialized software to analyze genetic and proteomic data with high costs. In addition, invasive sampling, such as the one used to collect aqueous humor samples, may not be practical in a clinical setting and for widespread use. Further work into non-invasive biomarkers is also necessary using imaging techniques improving their performance to detect early structural changes. Advancement in imaging techniques coupled with AI may provide higher accuracy and affordable solutions to implement into myopia clinics in a near future.”
- What are the technical and economic limitations and ease of use?
We added some thoughts on lines 876-883.
“Current analysis of biomarkers relies on advanced laboratory assessed with specialized software to analyze genetic and proteomic data with high costs. In addition, invasive sampling, such as the one used to collect aqueous humor samples, may not be practical in a clinical setting and for widespread use. Further work into non-invasive biomarkers is also necessary using imaging techniques improving their performance to detect early structural changes. Advancement in imaging techniques coupled with AI may provide higher accuracy and affordable solutions to implement into myopia clinics in a near future.”
Unfortunately, the current review does not provide the answers.
Response: We appreciate the comments and suggestions from the reviewer. We have addressed these questions in the Discussion section and have provided answers.
Reviewer 4 Report
Comments and Suggestions for Authors
This review article presents the current research on biomarkers and myopia, highlighting the challenges and opportunities in biomarkers research for myopia. It is well written and reviewed. However there some question should be clarified.
In table 1, “Smaller in myopic eyes” and “Thicker in myopic eyes: should be described more clearly.
The definition of high myopia and pathologic myopia should be addressed. Is the same definition for all studies listed in table1 and 2?
Author Response
Response to Reviewer 4 Comments
Comments and Suggestions for Authors
This review article presents the current research on biomarkers and myopia, highlighting the challenges and opportunities in biomarkers research for myopia. It is well written and reviewed.
We appreciate the comments and suggestions from the reviewer.
However there some question should be clarified.
Comment 1: In table 1, “Smaller in myopic eyes” and “Thicker in myopic eyes: should be described more clearly.
Response 1: We have added more details as much as the data are available for each included article.
The definition of high myopia and pathologic myopia should be addressed. Is the same definition for all studies listed in table1 and 2?
Response 2: We thank the reviewer for the suggestion. We have added the details of myopia definition in both Table 1 and 2.
Round 2
Reviewer 3 Report
Comments and Suggestions for Authors
In my opinion, the manuscript is now ready for publication.
Reviewer 4 Report
Comments and Suggestions for Authors
no more comment